

# A study of trust mining algorithms for beacon nodes in large-scale network environments

Yanyan Jiang

School of Information Engineering and Internet of Things, Huzhou Vocational & Technical College, Huzhou, China

## ABSTRACT

In a large-scale network environment, node positioning is prone to large deviations. Mining beacon node trust is the basis for precise node positioning in the network environment. Therefore, this article studies the trust degree mining algorithm of beacon nodes in a large-scale network environment. First, according to the distance error evaluation and probability function of beacon nodes in the large-scale network environment, the direct trust degree of beacon nodes is obtained. The trust degree is converted into influence, and the influence of beacon nodes is mined using the seepage theory to determine the beacon node with the highest impact in the large-scale network environment. Then, according to the influence of nodes, received signal strength indicator (RSSI) is used to optimize the conventional distance vector hop (DV-Hop) node location algorithm. The influence weights the average hop distance of beacon nodes. The weight of the influence of beacon nodes defines the average hop distance of unknown nodes. The average hop distance information of unknown nodes is taken from more high-influence beacon nodes, solving the problem of significant positioning errors caused by the uncertainty of location targets. Finally, the security status of nodes is reflected according to the degree of trust of different nodes to beacon nodes. The experimental results show that the algorithm can accurately locate other nodes in a wide network environment when the number of beacon nodes and communication distance change, and the trust degree of nodes mined can accurately reflect the security status of nodes.

# INTRODUCTION

In the ever-expanding wide-area network environment, the stability and reliability of beacon nodes, as the key elements of network localization and navigation, are crucial to the functioning of the entire network (*Shah, Kumar & Masud, 2023*; *Ng, She & Spachos, 2023*). The trustworthiness of beacon nodes, *i.e.*, the degree of the network's trust in the accuracy and security of its information, has become the core index of network performance and security (*Joshi & Sharma, 2023*). By carefully analyzing and deeply mining the trust level of beacon nodes, the network can improve its node operations' predictability and enhance its resilience against malicious attacks to provide network users with a more reliable service

Corresponding author
Yanyan Jiang, jyy55555@126.com

experience. However, with the increasing network scale and the complexity of the network environment, how to effectively mine and evaluate the trust degree of beacon nodes (*Sankar et al., 2023*; *Kalpana et al., 2024*) has become a hot spot and a difficult point in the current network technology research.

Relevant scholars have studied the degree of trust and the mining of beacon nodes. For example, in *Nadeem & Azeem (2023)*, all nodes in the network are located by selecting a specific node (beacon set). If the resolution (beacon node) set is fault-tolerant, it is also called a fault-tolerant beacon set. The minimum cardinality of this beacon set is called the fault tolerance metric dimension of the graph, and the beacon node mining is completed according to the fault tolerance metric dimension in the new tag. However, the network is dynamic, and the trust degree of nodes may also change over time. This fault-tolerant beacon set method may not be suitable for the rapidly changing network environment because it may be challenging to promptly adapt to the changes in node trust. In *Kuriakose, Joshi & Bairwa (2023)*, a statistical model was constructed to mine malicious beacon nodes in *ad hoc* networks, and the specific impact of multiple attack modes on node location mechanism was analyzed in depth. This model not only reveals how malicious beacon nodes can covertly track mobile nodes but also discusses how these nodes can distort the communication link, interfere with the normal operation of the trilateral measurement method and the received signal strength indicator, and thus mislead the discovery process of unknown nodes. The model also highlights the complex challenges encountered using trilateral measurement technology to detect damaged beacon nodes. However, the trilateral measurement and the received signal strength indicator may have errors, affecting the positioning accuracy of unknown nodes and the detection of damaged beacon nodes. *Santhameena & Manikandan (2022)* proposed a group acknowledgement medium access control (GACK-MAC) mechanism for transmitting aggregated data frames to a single destination and effectively reduced the overhead in IEEE 802.15.4 beacon wireless sensor networks. In this network environment, star networking is adopted, in which the coordinator is responsible for communication coordination with all other member nodes. The system locates the nodes in the large-scale network environment according to the degree of trust of beacon nodes to realize the effective deployment of nodes. However, it is worth noting that in the star and peer tree topology, the coordinator undertakes the coordination task with all member nodes, which may lead to the centralized risk of trust evaluation. Once the coordinator is attacked or captured, the trust evaluation system of the entire network may face the risk of collapse. *Godi & Janakiraman (2022)* uses the node clustering technology of the border shepherd dog optimization algorithm in the *ad hoc* network to mine the node trust in the network to achieve the best cluster head selection and node positioning to minimize the network overhead in the case of uncertain node density. However, this method uses a centralized method to select cluster heads, which may make cluster heads become the main target of attacks. If the cluster head is attacked, the trust of the entire cluster may be questioned.

There are some limitations to several existing methods like distance vector-hop (DV-Hop) (*Shah, Kumar & Masud, 2023*), received signal strength indicator (RSSI) (*Joshi & Sharma, 2023*), and the malicious node detection models like *Kuriakose, Joshi & Bairwa (2023)*.

DV-Hop (*Shah, Kumar & Masud, 2023*) is also not efficient with accuracy in large-scale networks with frequently changing topology, and RSSI (*Joshi & Sharma, 2023*) is interfered with by other signals and affected by the environment. The approach presented in *Kuriakose, Joshi & Bairwa (2023)*, while valuable for identifying malicious nodes, does not have any mechanism of monitoring changing levels of trust in real-life topologies. Therefore, we introduce beacon node trust mining and percolation theory to improve both positioning precision and network security to better address these shortcomings.

Beacon node trust mining has significant advantages in data reliability, positioning effect and network security (*Zhang et al., 2022*). It effectively improves the accuracy and reliability of data transmission in the network environment and ensures the authenticity and integrity of information in the network. At the same time, by accurately mining the trust degree of beacon nodes, the positioning accuracy in the network environment is significantly improved, and the navigation ability of the network is enhanced. More importantly, trust mining can build a solid defense line for network security, effectively prevent malicious node intrusion and attack, ensure the security and stability of the network environment, and provide users with a more secure and reliable communication experience.

In the continuously growing wide-area network (WAN) environment where the usage of beacon nodes is obligatory for the network localization and navigation, their stability and reliability determine the work of the whole network. The credibility, that is, the accuracy and security of information within the beacon nodes, has become the focal point of network and security performance parameters. Therefore, it is possible to strengthen the predictability of beacon node operations as well as improve the network's ability to counteract malicious attacks by focusing on increasing the level of trust in beacon nodes through analyzing and enhancing its deep mining. However, existing network localization algorithms such as DV-Hop and RSSI may still be inadequate in such modern, complicated, and intricate network scenarios. However, with high error rates in large-scale networks and the inability to scale up or down according to the number of nodes, DV-Hop has a low-complexity algorithm in place. Similarly, the RSSI-based localization is also predominantly influenced by environmental interferences and hence imposes inexact positioning, particularly in places characterized by interference or multipath signals. These limitations make it challenging for traditional approaches to solving the medium access control (MAC) problem to facilitate accurate and secure localization in dynamically changing networks. However, our solution extends these methods by the use of beacon node trust mining. To enhance the nodes localization precision and make a far better protection against the unauthorized attacks, we are planning to assess the beacon nodes' credibility level and then apply percolation theory in the networks of today's enormous measures.

To ensure accurate node positioning and network security in WSNs, the following problems are major challenges. DV-Hop and RSSI localization techniques give inefficient results consequently in large-scale, dynamic networks where nodes fail to linger for a longer period. Interference and multipath effects from the environment add additional distortion to signal measurements, prevent the nodes from being accurately placed, and

also degrade the network performance. Furthermore, security relates to beacon node trustiness for localization as beacon nodes are of critical importance. By the hand of malicious or compromised beacon nodes, a false geographical location can result in severe consequences in that it affects the functionality and security of the beacon needed to construct the network. Thus, as the number of nodes increases, these malicious attacks intensify, and this makes the current system of beacon node verification and validation not very effective, thereby exposing problems of location estimation and security.

To overcome these challenges, this study proposes a beacon node trust mining algorithm based on the above analysis for a broad-range network environment. Using the probability value of the distance error case to calculate the trust degree of beacon nodes directly, the trust degree of beacon nodes is obtained, and then using the percolation theory, the trust degree of beacon nodes is mined to get the result of the maximum influence beacon nodes.

## BEACON NODE TRUST MINING

This study calculates the direct trust degree and converts it into influence by evaluating the distance error and probability function of beacon nodes. Use seepage theory to mine high-impact beacon nodes and optimize the DV-Hop location algorithm. By weighting the influence of beacon nodes and calculating the average jump distance, the accuracy of locating unknown nodes can be improved, thereby reducing errors caused by the uncertainty of the positioning object. At the same time, the network security status is reflected according to the degree of trust between nodes to enhance the security and reliability of the network.

### Node trust calculation calculation
#### *Trust calculation for beacon nodes*

Calculating the trustworthiness of beacon nodes in a large-scale network environment is critical. Considering the trust level of beacon nodes, the average hop distance of the target node can be calculated. At the same time, the trust level of beacon nodes can provide a basis for evaluating the node's security through the trustworthiness of any node to beacon nodes to react to the nodes in large-scale network environments whether or not the node suffers from an attack. The sample data set $\delta_{i,j} = \{\delta_1, \delta_2, \delta_3, \cdots, \delta_n\}$ of distance errors between the distance error of the trusted nodes $i$ $j$ and with the maximum distance error, the minimum distance error and the average distance error are $\delta_{\max}, \delta_{\min}, \delta_{avg}$, whose difference can describe the size of the possibility that this distance error is normal, uncertain and abnormal, the function is shown in Fig. 1. Based on the fuzzy set theory, the probability distribution function formula is:

$$P_T(\delta_i) = \begin{cases} \dfrac{\delta_i - \delta_{\min}}{\delta_{avg} - \delta_{\min}}, & \delta_{\min} \leq \delta_i \leq \delta_{avg} \\ 0, & \delta_{avg} \leq \delta_i \leq \delta_{\max} \\ 0, & \delta_i < \delta_{\min} \cup \delta_i > \delta_{\max} \end{cases} \quad (1)$$

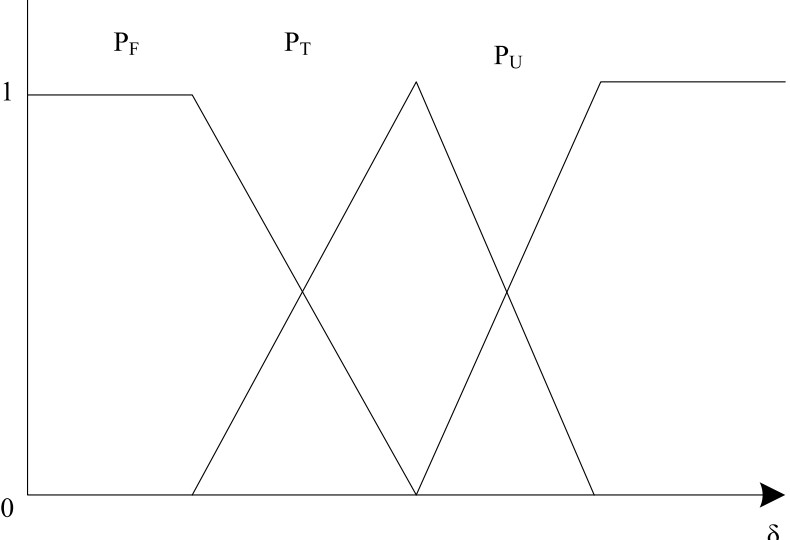

**Figure 1 Probability allocation function.**

$$
P_U(\delta_i) = \begin{cases} 0, & \delta_{\min} \leq \delta_i \leq \delta_{avg} \\ 1 - \dfrac{\delta_{\max} - \delta_i}{\delta_{\max} - \delta_{avg}}, & \delta_{avg} \leq \delta_i \leq \delta_{\max} \\ 1, & \delta_i < \delta_{\min} \cup \delta_i > \delta_{\max} \end{cases}
\tag{2}
$$

$$
P_F(\delta_i) = \begin{cases} 1 - \dfrac{\delta_i - \delta_{\min}}{\delta_{avg} - \delta_{\min}}, & \delta_{\min} \leq \delta_i \leq \delta_{avg} \\ \dfrac{\delta_{\max} - \delta_i}{\delta_{\max} - \delta_{avg}}, & \delta_{avg} \leq \delta_i \leq \delta_{\max} \\ 0, & \delta_i < \delta_{\min} \cup \delta_i > \delta_{\max} \end{cases} .
\tag{3}
$$

Among them, $P_T(\delta_i)$, $P_U(\delta_i)$, $P_F(\delta_i)$ represent the probability values that each distance error is normal, uncertain and abnormal, respectively. When $\delta_{\min} \leq \delta_i \leq \delta_{avg}$ the distance error is called an average distance error, when $\delta_{avg} \leq \delta_i \leq \delta_{\max}$ it is called an uncertain distance error, when $\delta_i < \delta_{\min} \cup \delta_i > \delta_{\max}$ it is called an anomalous distance error.

The results of the direct trust assessment are expressed in terms of the level of direct trust $B$. Let the beacon node $L_i$ conduct a direct trust evaluation of any target node $L_u$ in an extensive range of network environments, then the direct trust of $L_i$ $L_u$ this:

$$
B_{i,u} = f_{T_{i,u}}(\delta_{i,u}), f_{F_{i,u}}(\delta_{i,u}), f_{U_{i,u}}(\delta_{i,u}).
\tag{4}
$$

Among them, $f_{T_{i,u}}(\delta_{i,u}), f_{F_{i,u}}(\delta_{i,u}), f_{U_{i,u}}(\delta_{i,u})$ denote the probability value for the beacon node $L_i$, considering the target node $L_u$ is credible, implausible, and uncertain, and their sum is 1.

### Influence-based trust mining of beacon nodes

In trust mining, trust is converted into influence; the more significant the trust of a beacon node, the greater the influence of that beacon node, and influence maximization aims at identifying, in a wide range of network environments $k$, a beacon node, such that through this $k$ beacon node produces the maximum influence spreading range under the influence spreading mechanism. The idea of percolation theory is adopted to mine the nodes of the most influential beacons. It is assumed that the degree of importance of beacon nodes is positively correlated with the influence of nodes, so the joint propagation of influence between beacon nodes is ignored. Based on the see page theory $k$, individual influence maximization beacon node mining avoids the shortcomings of traditional mining algorithms to a certain extent and not only considers the degree of node importance but also introduces the concept of joint influence intensity, which considers the joint influence factors among beacon nodes.

Let the vector $g = (g_1, g_2, \cdots, g_n)$ be a labelling vector representing whether a beacon node in the network is an influential node or not, where:

$$g_i = \begin{cases} 0, & n_i = removed \\ 1, & n_i = present \end{cases} \tag{5}$$

The proportion of influential nodes in the network is:

$$q \equiv 1 - \langle g \rangle. \tag{6}$$

The percolation theory views the influence-maximizing beacon node mining process as the continuous removal of "super-influence" nodes from the network nodes so that the influence can not be spread in the network, which $q_i$ is the proportion of beacon nodes removed from the network. Let $G(q)$ be the average probability of no affected nodes in the network after the influence diffusion behaviour ends in a network with $q$ "super influence" nodes. Set $v = (v_1, v_2, \cdots, v_n)$, of which $v_i$ is the probability of node $i$ final unaffected, *i.e.*, when $t \to \infty$ beacon nodes $i$ are the probability of an inactive state, the expression for $G(q)$ is:

$$G(q) = \frac{\sum_{i=1}^{n} v_i}{n}. \tag{7}$$

Thus, the influence maximizing beacon node mining can be transformed into an optimal percolation problem to find the "super influence" nodes with optimal $q_i$ proportion (removed nodes), minimizing the probability $G(q_i)$ of unaffected nodes in the final network, the mathematical formalization of the problem is expressed as follows:

$$q_i = \min\{q \in [0, 1] : \min G(q)\}. \tag{8}$$

When $q \geq q_i$ a series of influence joint diffusion nodes exist in the large-scale network environment, influence is diffused from these nodes to the whole network when $q \leq q_i$ it suggests that small, isolated localized worlds in the wider network environment prevent influence from spreading to the entire network.

To measure the actual influence of a node in the network, consider removing the node virtually and examine the transfer of influence before and after the removal of the node. Let two nodes $i\ j$ and introduce markers $v_{i \to j}$ to represent the probability that the node $i$ is ultimately not affected when a node $j$ is virtually removed. The mathematical formalization of the local influence diffusion tree centred on the node $i$ is expressed as follows:

$$v_{i \to j} = n_i \left[ 1 - \prod_{k \in \frac{i}{j}} \left(1 - \omega_{k \to i} v_{k \to i}\right) \right] \tag{9}$$

$$\omega_{k \to i} = \frac{\sum_{t \in \frac{i}{j}} IS(\lambda_k, \lambda_t)}{|k| - 1} \tag{10}$$

where the beacon node $i$ itself is an influence node, i.e., $g_i = 0$ obviously $v_{i \to j} = 0$, when the node $i$ is a non-influential node, i.e., $g_i = 1$, the probability of whether the node is finally activated or not is related to the neighbouring nodes. Among them, $\omega_{k \to i}$ the local average influence diffusion coefficient of the node $k$ when the neighbour node $i$ of node $\omega$s is virtually removed $\frac{i}{j}$, the collection of neighbour nodes of the node $i$ after the node $j$ is virtually removed $\lambda$, and the influence of the nodes.

In the above local influence diffusion model, it can be verified that it $i \to j\ i\ j \in n$ $\{v_{i \to j} = 0\}$ is a global stabilization solution for all. Construct $2M \times 2M$ a system of closable equations to solve the global optimum $g$, introducing the linear operator:

$$\varpi_{k \to o, i \to j} = \frac{v_{i \to j}}{v_{k \to o}} \Big|_{v_{i \to j} = 0} \tag{11}$$

Among them $\varpi$ is the linear operator defined in the local diffusion model of directional edge influence in $2\varpi$ bars, which can be expressed by the $2M \times 2M$ matrix based on the non-regression matrix. Therefore, after introducing the influence diffusion coefficient into the diffusion model, defining the weighted non-regression matrix into the diffusion model:

$$\varpi_{k \to o, i \to j} = g_i \omega_{k \to i} \beta_{k \to o, i \to j}. \tag{12}$$

Among them $\beta$ is a non-regressive matrix.

The percolation theory transforms the influence-maximizing node mining problem into an influence diffusion matrix $\varpi$ to solve the problem of maximum eigenvalue. According to Perron Frobeniu's theorem, the maximum eigenvalue of the matrix $\omega\beta$ is strictly less than the function of the matrix $\omega\beta$, i.e., when $\varpi \leq \omega\beta$ (the sizes of the elements in the same corresponding positions of the two matrices strictly satisfy the inequality

requirement) there is $\lambda(\varpi) < \lambda(\omega\beta)$. Defining the problem by finding the globally optimal sequence $g^*$ of the influential node labels $\lambda(g^*, q)$ minimizes the problem, *i.e.*,

$$\lambda(g^*, q) \equiv \min_{g\langle g\rangle = 1-q} \lambda(g, q). \tag{13}$$

Among them $q$ is the ratio of beacon nodes to nodes in a wide-range network environment. Solving Eq. (13) yields $B'$ the highest-impact beacon node in the large-scale network environment.

## Trust-based node localization for large-scale network environments

After mining the most influential beacon node using the percolation theory, the trust degree of this beacon node is used as the basis to locate other nodes in the wide-range network environment, thereby enabling the accurate positioning of target nodes in the wide-range network environment.

### *Network wireless network node location based on the DV-HOP algorithm*

The positioning process of the DV-Hop algorithm (node positioning algorithm in wireless sensor networks) is as follows (*Mohanta & Kumar Das, 2022*):

Step 1: Broadcasting location. A beacon node broadcasts a packet of data containing its coordinate information to its neighbouring nodes in the sensor network in the form of flooding, including {coordinates $(x, y)$, node number *id*, the minimum number of hops *Hop*}, where the minimum number of hops *Hop* initialized to 0. The receiving node records the packet of each arriving beacon node and saves only the packet with the minimum hop count, then adds 1 to the minimum hop count and forwards it to its neighbouring nodes; during this process, the receiving node ignores packets from the same beacon node that contain larger hop counts. Through this stage, all the network nodes can record the minimum hop count packet information to each beacon node (*Bai et al., 2024*).

Step 2: By calculating the average hop distance of beacon nodes in the network, the average hop distance of each beacon node in the network is determined based on the position data and the shortest hop count of other beacon nodes obtained in the first stage. This calculation process follows the following formula:

$$HopSize_i = \frac{\sum_{j \neq i} \sqrt{(x_i - x_j)^2 + (y_i - y_j)^2}}{\sum_{j \neq i} h_{ij}}. \tag{14}$$

Among them. $(x_i - y_i)$, $(x_j - y_j)$ denotes the coordinates of the beacon node $i, j$ $h_{ij}$ and indicates the minimum number of hops between the beacon node $i$ and $(i \neq j)$ (*Kaur, Gupta & Mittal, 2022*).

Step 3: Coordinate calculation. The beacon node broadcasts the calculated network average hop distance to the whole network; each unknown node only records the first network average hop distance value, forwards it to the neighbouring nodes, and discards

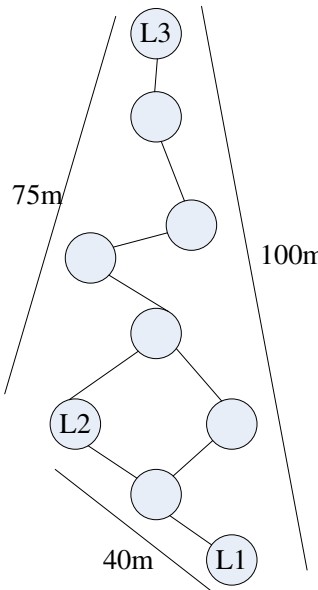

**Figure 2  DV-Hop localization process.**  

the subsequent received values; this distribution strategy ensures that the majority of unknown nodes only receive the network average hop distance value of the nearest beacon node. When the unknown node receives the hop distance value, it uses this value as its distance per hop in the network and combines it with the minimum number of hops of each beacon node obtained from the first stage records to calculate the distance to each beacon node. When the unknown node gets the coordinates of three or more beacon nodes, it calculates its coordinates using trilateral or great likelihood measurements (*Mohanta & Kumar Das, 2022*).

The positioning implementation process of the DV-Hop algorithm is shown in Fig. 2. In Fig. 2:

Phase I: Beacon nodes *L*1 *L*2 *L*3 in the network broadcast their coordinate information by flooding; after all nodes in the network receive the broadcast packet, the packet information of the beacon node is retained according to the minimum hop count rule.

Phase II: *L*1 *L*2 *L*3 Based on the minimum number of hops and coordinate information received from each other in the first stage, their network average hop distance is calculated using the formula (network average hop distance = total distance/total number of hops). The results are then broadcast to the network.

Phase III: The unknown node *u* receives only the average hop distance value of the first arriving network as its network per-hop distance. We assume that the average hop distance of the network is obtained $\frac{(40+75)}{(2+5)} = 16.4$ for the unknown node *u* from *L*2 the first. The distance between *u* the three beacon nodes *L*1 *L*2 *L*3 *L*1 *L*2 *L*3 is 49.2, 32.8, and 49.2, respectively. Finally, the coordinate positions of the unknown node *u* can be calculated by the trilateration measure (*Rayavarapu & Mahapatro, 2024*).

### Improvement of DV-Hop algorithm based on node influence and RSSI

In the standard DV-Hop algorithm, the hop count information between nodes is too dependent when calculating the estimated distance between nodes. However, in a large-scale network environment, sensor nodes are numerous and distributed irregularly, so it is inappropriate to use Euclidean distance to represent the communication distance between nodes. The average hop distance obtained by dividing the sum of Euclidean distances between nodes by the sum of hops has errors. However, when calculating the distance between beacon nodes and unknown nodes, the average hop distance with errors is multiplied by the corresponding hops, leading to error accumulation. As a result, the more hops between nodes, the greater the distance estimation error and the lower the positioning accuracy. The proposed algorithm incorporates trust mining as an additional feature to the node localization process by assigning trust values to the beacon nodes by the distance errors described by the fuzzy set theory's normal, uncertain, and abnormal membership values. These trust values are used to sort the beacon nodes, to select a higher trust node to be processed to maximize the influence to propagate across the rest of the network. For node localization, the DV-Hop algorithm is used, and the estimates of trust values for beacons are included in the hop distance determination. The nodes with higher trust play a major role in the localization process, thus increasing the reliability of the result achieved. The integration of trust mining with influence maximization improves the node localization techniques, decreasing the chances of inaccurate localization and improving the overall reliability and results of the network.

① Node location based on received signal strength indication (RSSI) technology

The attenuation of wireless signal strength is approximately logarithmic to the transmission distance. The signal transmission model is a model for fitting multiple groups of measured signals and distance to convert signal strength and distance (*Carpi et al., 2023*). The range conversion accuracy affects the final positioning result, so establishing the transmission model needs to be very accurate. Standard signal transmission models include the free space transmission model, the Shadowing propagation model, *etc*. The commonly used Shadowing propagation model is:

$$S^{RSSI}(d) = S^{RSSI}(d_0) - 10n \lg\left(\frac{d}{d_0}\right) + X_\sigma \tag{15}$$

of which $d$ is the power for distance $S^{RSSI}(d)$ $d$, *i.e.*, the signal strength. $d_0$ is the reference distance. $S^{RSSI}(d_0)$ is the reference signal strength. $n$ is the signal attenuation factor; $X_\sigma$ it is the random noise obedience distribution $(0, \sigma^2)$. In general, $d_0$ it takes the value of 1, *i.e.*, the signal strength at 1m from the beacon node is used as a reference value, denoted as $\eth$, then Eq. (2) can be simplified as follows:

$$S^{RSSI}(d) = \eth - 10n \lg d \tag{16}$$

Formula (3) is the commonly used signal transmission model, which realizes the conversion of measured signal strength and distance. To avoid the RSSI error and the

extreme value in case of emergencies as much as possible, the average value of multiple received signals is generally adopted, which can reduce the RSSI error as much as possible (*Islam et al., 2023*).

② Optimization of DV-Hop algorithm

Consecutive hops: The ratio of the distance between two nodes within each other's communication range to the communication radius. Let the communication radius of the node $R$, the distance between the receiving node $j$ and the transmitting node $i$ is $d$, then the number of consecutive jumps is:

$$h_{ij} = \frac{d}{R}. \tag{17}$$

To find the number of consecutive hops of nodes, one needs to know the distance because the beacon nodes have specific coordinates. Still, the coordinates of the unknown nodes are unknown, so the number of consecutive hops between nodes is divided into two categories (*Kumar, Batra & Kumar, 2023*); the first is the number of consecutive hops between nodes when the neighbouring nodes are beacon nodes and the second type of consecutive hops between nodes when the neighbouring nodes have unknown nodes. In the following, these two types of consecutive hops are analyzed.

The first category: The number of hops between nodes when neighbor nodes are beacon nodes. When the neighbour node is a beacon node, the number of consecutive hops between the nodes is used in the distance between the nodes, and the Euclidean distance between the nodes' consecutive hops formula is:

$$d_{ij} = \sqrt{\left(x_i - x_j\right)^2 + \left(y_i - y_j\right)^2} \tag{18}$$

$$h_{ij} = \frac{d_{ij}}{R}. \tag{19}$$

The second category: Hops between nodes when neighbor nodes have unknown nodes. Since the coordinate information of unknown nodes is unknown, it is necessary to introduce the Shadowing model in RSSI technology, use the Shadowing model to calculate the measurement distance between nodes, and use the measurement distance combination Formula (4) to calculate the hops between nodes (*Pinto & Oliveira, 2024*). The positioning accuracy of nodes has been improved by improving the hop number between nodes. Still, simultaneously, the average hop distance information of unknown nodes only comes from one beacon node. In this way, in the positioning process, if an emergency causes a serious error in the average hop distance of beacon nodes, it will affect the average hop distance of unknown nodes, thus causing an increase in node positioning error. To avoid this problem, unknown nodes' average hop distance information is taken from as many beacon nodes as possible (*Wang & Song, 2023*).

For the problem of the average hopping distance of unknown nodes, the influence of beacon nodes is introduced. The influence is used to give weight to the average hopping distance of beacon nodes, and the weight of the influence of beacon nodes is used to define

the average hopping distance of unknown nodes (*Meng et al., 2023*; *Sun et al., 2020a,* *2020b*; *Zhu et al., 2024*; *Luo et al., 2022*; *Gong et al., 2024*).

Assuming a large-scale networked environment has $N$ beacon node, using the hop information of nodes to obtain the hop matrix $H$ between $N$ beacon nodes, taking the original algorithm, to find the average hop distance of each beacon node $Hopsize_i$. Then, find the measured distance $\hat{d}_{ij}$ between the beacon nodes $i$ and beacon nodes $j$ with the formula:

$$\hat{d}_{ij} = Hopsize_i \cdot h_{ij} \tag{20}$$

The average error per hop is:

$$hopdiff_{ij} = \frac{abs\left(d_{ij} - \hat{d}_{ij}\right)}{hop_{ij}} \tag{21}$$

Combining the error with the node influence, the average hopping distance of the unknown node was found to be:

$$Hopsize = \sum_{i=1}^{m} B'_i \times Hopsize_i \tag{22}$$

In the formula, $B'_i$ it is the highest-impact beacon node in a wide-ranging network environment.

The node influence degree obtained from the calculation can be brought into Eq. (22) to enhance the node localization accuracy in two ways:

The first aspect, the number of hops between nodes, is defined as the number of consecutive hops so that the number of hops between nodes can accurately reflect the distance.

The second aspect is to calculate the average per-hop error of the beacon node, weigh the average hop distance of the beacon node, and obtain the average hop distance of the unknown node by averaging and summing.

# EXPERIMENTS AND ANALYSIS OF RESULTS

## Experimental subjects

The automatic irrigation system in an industrial park is studied as an example. Figure 3 shows the layout of the park's greening area.

Table 1 details key parameters such as park size, green percentage, and irrigation equipment. As can be seen in Table 1 below, parameters were chosen as they affect the efficiency of the irrigation system as well as the performance of the sensor network. The park area (100 hectares) and the afforested area (40 hectares) dictate the size of the park and, by extension, water distribution and sensor node distribution. The green percentage is relevant as 40% since regions with more vegetation demand more water and thus affect the irrigation plan. An automatic sprinkler system is used to guarantee efficient and automated water delivery The number of irrigation equipment units (500 units) is enough to cover the

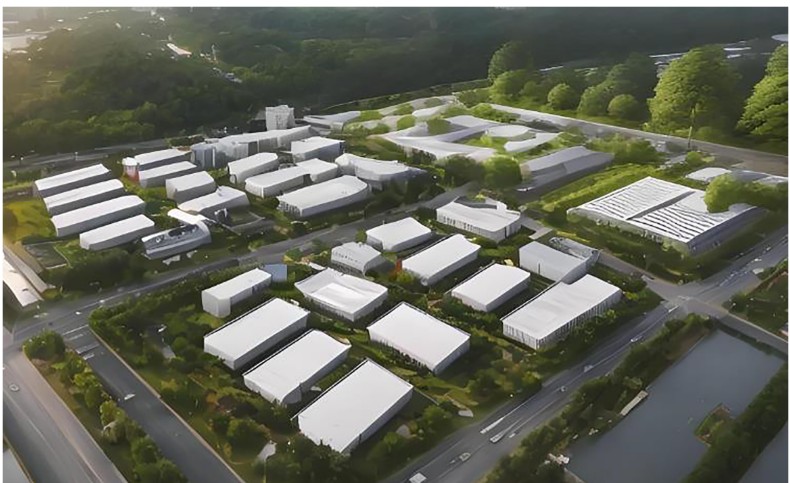

**Figure 3  Overall image of the industrial park.**

Table 1  Park parameters.

| Attribute | Parameter |
|---|---|
| Park area | 100 hectares |
| Afforested area | 40 hectares |
| Number of green zones | 5 |
| Type of irrigation equipment | Automatic sprinkler irrigation system |
| Quantity of irrigation Equipment | 500 |
| Water source type | Groundwater/Rainwater |
| Average daily water supply | 1,000 m$^3$ |
| Control mode | Wireless Sensor Network |
| Control center | Park Management Center |

park area and the green areas. As the parameters defining borehole and rainwater availability for irrigation, the water source type and average daily water supply per 1,000 m$^3$ were used. Finally, the wireless sensor network as the control mode and park management center as the control center allow real-time control for irrigation and performance of the wireless sensors.

Table 2 demonstrates the parameters of the wireless sensor network for controlling the irrigation system, including the number of nodes, communication range, and transmission frequency, which ensures intelligent irrigation and efficient use of water resources.

## Experimental data

Trust mining is performed on the network beacon nodes of a green partition in the park, and the topology of the wireless sensor network part of this green partition is shown in Fig. 4.

The trust mining results for different nodes are shown in Table 3.

**Table 2 Parameters of large-range wireless sensor networks.**

| Attribute | Parameter |
|---|---|
| Total number of nodes | 1,000 |
| Number of beacon nodes | 50 |
| Ordinary sensor node | 950 |
| Node communication range | 150 m |
| The average lifespan of nodes | 3 year |
| Data transmission frequency | 15 min |
| Network coverage area | 120 hectares |
| Management software | Irrigation system |

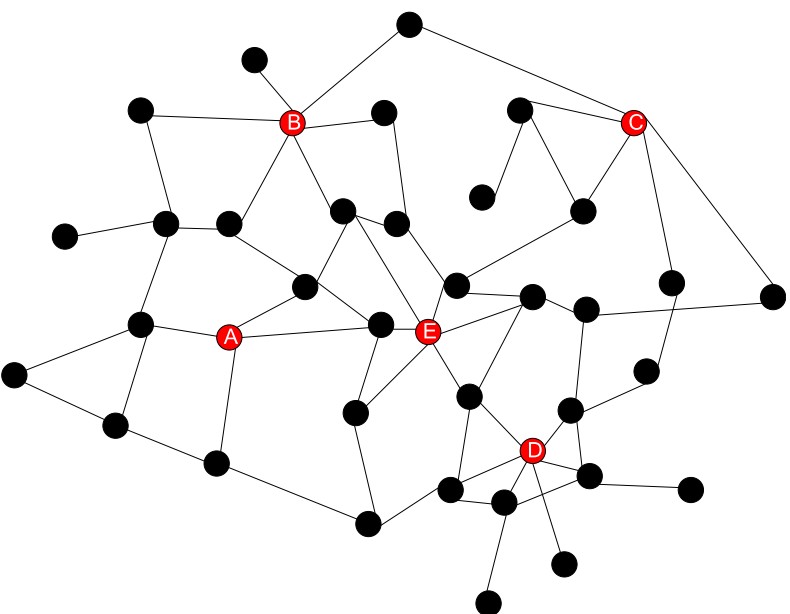

**Figure 4 Partial network node topology structure.**

**Table 3 Trust mining results of beacon nodes.**

| Node number | Trust level |
|---|---|
| A | 0.62 |
| B | 0.57 |
| C | 0.55 |
| D | 0.64 |
| E | 0.72 |

Figure 4 and Table 3 show the distribution of beacon nodes and the trust degree of beacon nodes A, B, C, D and E. Each beacon node plays a different role in the wireless sensor network, and its trust reflects its reliability and influence in the positioning process.

Specifically, in Fig. 4, it can be observed that node E is in a key position, consistent with the highest trust in Table 3. The beacon node E, with the highest trust, will play a decisive role in locating other nodes because its signal is stable and its coverage is wide. Using beacon node E can more accurately locate the position of other nodes. In our network, beacon nodes A, B, C, D, and E are important since they are used for positioning and help to perform a trust calculation. These nodes are placed predictively to guarantee reliable localization and support neighboring nodes' trust estimates. The nodes chosen are based on location distribution, stability of the node, and its capability to steer the network. These are placed based on coverage to ensure that every node within the network has a chance to communicate with a beacon. Selection is also determined by its stability and location in the middle of the network, making it easy to radiate the trust information. Moreover, the influence of beacon nodes is optimized using the influence maximization approach, limiting the exposure to attacks to achieve higher security. Beacon node placements allow the network to remain stable despite some nodes being inactive or having certain levels of interference. For instance, since node E enjoys a high local trust value and is located at the network's center, it is solely responsible for the correct localization of other nodes in the immediate neighbourhood set it hosts; its presence guarantees the accurate operation of the network's position estimation algorithm.

To verify the positioning of network nodes based on beacon node trust in a large-scale network environment, the standard RSSI algorithm and the standard DV-Hop algorithm are compared with the algorithm in this article. The results are shown in Fig. 5. Our method, in contrast with the DV-Hop algorithm, considers only hop counts and supposes a uniform distribution of nodes. Still, this one uses the trust values of beacon nodes and can assign them more influence in the localization process. This decreases the compound error for distance estimation by a great deal. On the other hand, the RSSI method benefits from performance enhancement by signal strength measurements. Nevertheless, the technique is impacted by environmental noise, interferences, and varying signal strength. However, the trust degree is not considered in the DV-Hop algorithm, which demerges this proposed algorithm and has incorporated the trust degree with improvements on the structure and accuracy of the DV-Hop algorithm in dynamic and large-scale networks. It can be seen from Fig. 5 that the normalized positioning error reduces sharply as the number of beacon nodes increases. In all cases, the proposed method outperforms the other methods.

After analyzing the data in Fig. 5, it is evident that the positioning accuracy of the traditional DV-Hop algorithm in wireless sensor networks has some limitations. Although the positioning error decreases with the increase in the number of beacon nodes, the positioning error is still as high as 0.3 when there are 50 beacon nodes, which could be better for situations requiring high-precision positioning. In contrast, the RSSI algorithm shows certain advantages under the same conditions, reducing the positioning error to about 0.17, which shows its effectiveness in improving positioning accuracy. However, the algorithm proposed in this article significantly improves when there are 50 beacon nodes, and the positioning error is only 0.1. This breakthrough benefits from the algorithm's in-depth consideration of location and beacon nodes' trust degree. This algorithm

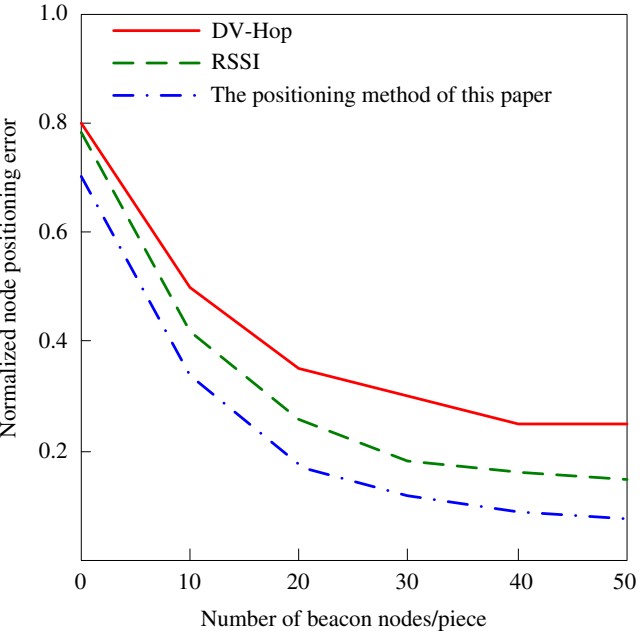

**Figure 5 Normalized positioning error of different positioning methods.**

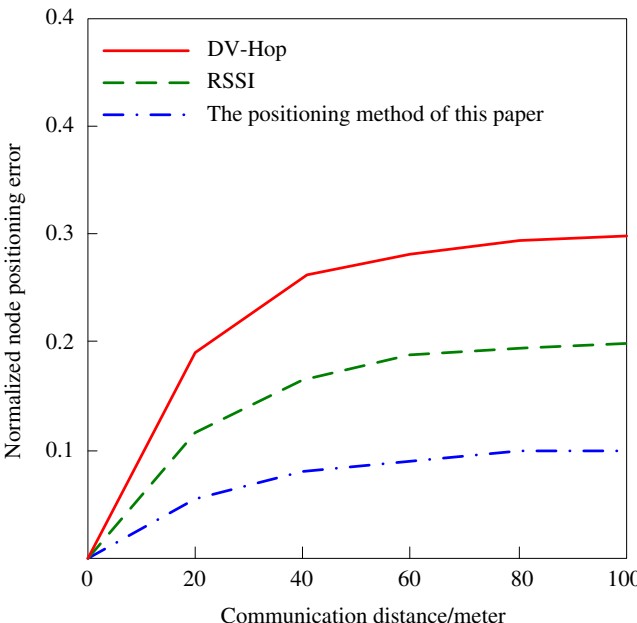

**Figure 6 The influence of communication distance on node positioning error.**

significantly improves location accuracy by introducing the trust degree weighting mechanism.

Communication distance is also a key factor influencing the node localization effect. In the experiment, the localization errors of different algorithms are observed by changing the

communication distance between the beacon node and the target node, and the results are shown in Fig. 6. Earlier methods, such as DV-Hop, use hop count in the route that has errors compounded with distance in the communication methods. Likewise, the RSSI method is devised with the inaccuracies of environmental noise or signal attenuation because of longer distances, or in other words, it has poorer distance accuracy due to attenuation. On the other hand, the proposed algorithm avoids such challenges by incorporating trust values to scale the effect of beacon nodes. Nodes are ranked by their level of trust in the network, and because anchors have long-lived accurate distance measurements, they are given higher weights for localization. This means that more prominent nodes are reliable to put up with the other nodes that may give imperfect or irregular distances. By accurately selecting which nodes are trustworthy, the trust-based mechanism is a tool that keeps positioning errors low even in large, scalable and rapidly changing network environments as the communication distance grows. This optimizes on distance, hop count and signal reliability, which justifies the desirable results observed in Fig 6.

It can be seen from Fig. 6 that the communication distance will have an impact on node positioning. Since the DV-Hop algorithm is based on the number of hops, the positioning error will increase significantly with the distance increase. When the distance between the beacon node and the target node reaches 100 m, the error of this algorithm reaches 0.3, RSSI algorithm locates the node's power. Compared with the DV-Hop algorithm, the accuracy of the RSSI algorithm is slightly higher. When the communication distance is 100 m, the positioning error of the node is 0.2. As node density increases or beacon nodes are placed farther apart, the trust mechanism dynamically prioritizes high-trust nodes. By weighting reliable nodes more heavily, the algorithm reduces distance estimation errors and maintains positioning accuracy, ensuring scalability and robustness even in sparse or irregular network deployments. The algorithm in this article has the best effect. Because the node credibility is considered and the node is weighted, it can ensure the high-precision positioning of the beacon node and the communication node within the maximum communication distance. When the communication distance is 100 m, the node positioning error of the positioning algorithm in this article is only 0.1.

The trustworthiness of a beacon node is calculated using a fuzzy set theory-based model, which classifies distance errors into three categories: standard, open to question, and abnormal. The probability distribution function formula considers the distances of errors of the beacon node relative to target nodes—whether these are normal or symptomatic of an attack or a failure. Probability of uncertain errorTrust degree will be equal to the probability of normal error, probability of uncertain error, and probability of abnormal error. Ifies distance errors into three categories: normal, uncertain, and abnormal. The probability distribution function formula considers the distance errors between the beacon node and target nodes, determining whether the errors are normal or indicative of an attack or malfunction.

The formula used for the trust calculation is:

$$\text{Trust Degree} = \text{Probability of Normal Error} + \text{Probability of Uncertain Error} + \text{Probability of Abnormal Error}$$

Accordingly, to calculate the trust degree based on these probabilities, we exclude the possibility of beacon nodes that exhibit unstable or unreliable behaviour to obtain the high trust degree. Thus, compared with the current models that use reputation or direct observation methods (the Bayesian approach or evaluating direct trust), the proposed method integrates fuzzy set theory for error classification and percolation theory for influence maximization. This combined feature guarantees that the values of trust are also oriented to the credibility of beacon nodes and their impact on the overall social graph. Our approach extends simple trust calculation by adding joint influence intensity of involved beacon nodes, establishing a safer system for large-scale networks. This approach eliminates the weaknesses spotted in older methods of trust determination in which trust values are dependent on previous interactions or simple mathematical values. Our method achieves better performance and is more secure than the other methods because it concentrates on the reliability of beacon nodes and reducing the effects of a malicious node, which is more of a problem in large-scale WSNs.

The trust mining results of beacon nodes can also reflect the security of beacon nodes and corresponding target nodes; by understanding the actual situation to set the security threshold, nodes lower than the security threshold prove that the node may be in danger and should be subjected to malicious attacks. The results of randomly selecting 10 nodes to attack and calculating the trust degree of the node are shown in Table 4. First, nodes with trust scores less than the defined security threshold are excluded temporarily from the localization process because they might continue to disturb the process, while neighboring trusted nodes are compensated for their duty to stabilize the network. Second, the system quietly and proactively scans each isolated node. In a similar way, an adaptive recovery mechanism is implemented after a node has proved that it has not produced multiple failures, and if the node has performed a number of reliable responses, including low-distance errors, facilitation of stable communication, *etc.*, then a gradual trust recovery takes place. This approach leads to a denial of control and disruption of the network to the attacker, enables re-admission of the malicious nodes when appropriate, and provides correct spatial location and security of the network even under attack.

It can be seen from Table 4 that after the application of the algorithm in this article, the trust degree of nodes generally exceeds 0.9 without being attacked, which indicates that under normal circumstances, the reliability and stability of nodes are very high. However, once a node is attacked, its trust will decline significantly. Notably, the trust degree of beacon node 7 drops to the lowest level, only 0.11. This significant change reveals that the node may have suffered malicious attacks. It can be seen that trust is an effective indicator that can timely reflect whether the node has been attacked. The system proposed for beacon nodes efficiently identifies and prevents many typical attacks on the network to maintain stability and security. Regarding the DoS attack, it targets important shifts in

**Table 4 Trust level of nodes after being attacked.**

| Node number | Is it an attack | Trust level |
|---|---|---|
| 1 | Yes | 0.33 |
| 2 | Yes | 0.28 |
| 3 | Yes | 0.37 |
| 4 | No | 0.93 |
| 5 | Yes | 0.26 |
| 6 | Yes | 0.15 |
| 7 | Yes | 0.11 |
| 8 | No | 0.95 |
| 9 | Yes | 0.32 |
| 10 | Yes | 0.21 |

beacon node behaviour resulting from resource depletion due to large traffic. For Sybil attacks in which the attacker creates multiple fake identities, the trust mining algorithm identifies behavioural patterns, including irregular distance errors and communication pattern irregularities. Further, the false data-injected problem, where nodes send fake distance or coordinate data, is solved through the trust-based weighting mechanism that favours nodes accidentally that are most reliable in a consistent manner. In this case, monitoring trust levels and global directions in addition to such deviations as a drop in trust scores helps the system to determine the correct positioning and organization's failure-proof existence in case of active interference by unauthorized individuals.

Through the monitoring and change analysis of node trust, we can quickly identify the abnormal situation in the network and take corresponding measures to ensure the operation safety of the large-scale network. This method provides efficient network security management and helps maintain network stability and data security. As shown in the current work, the use of the proposed algorithm shows a high success rate when used in the simulation environment, and as stated, the approach involved is versatile to fit different real-life network conditions. Some of the additional environments for validation include environments with signal interference, such as urban environments, complex channel environments, such as industrial environments, and irregular node deployments, such as wooded environments. These environments bring in diverse challenges, including multipath effects, dynamic node density and obstacles in the environment to further challenge the algorithm. Thus, applying the trust-based weighting and the influence maximization, the algorithm can adapt to these challenges and keep superior positioning accuracy. More work in this case will focus on performing more simulations with these scenarios in order to confirm the realistic generalization capacity and efficiency of this algorithm.

Future work will apply machine learning (ML) techniques to improve the algorithm's efficiency. Through trust degradation patterns, supervised models such as Random Forest can identify malicious nodes, and RNNs or LSTMs can predict the next trust trends for preemptive isolation of nodes. The trust weights will be learned dynamically to reinforce

the localization in reinforcement learning, as unsupervised methods like K-means can be adjusted according to the node density to achieve better scalability and robustness in the dynamic network.

## CONCLUSION

In this article, we study an advanced algorithm based on the trust mining of beacon nodes in various network environments. The algorithm can not only realize accurate positioning of target nodes but also effectively judge the security status of target nodes by deeply mining the trust relationship between beacon nodes and any target nodes to be located. This trust mining technology provides a new perspective for network monitoring and management and greatly improves the security performance and reliability of the network. In the future, with the continuous progress of technology, we expect to develop more efficient mining methods, such as combining machine learning and artificial intelligence techniques, to realize more intelligent and adaptive trust evaluation. In addition, exploring how to update and optimize trustworthiness models in real-time in dynamic network environments, as well as how to extend trustworthiness mining algorithms to more types of network applications, such as the Internet of Things (IoT) and in-vehicle networks, *etc.*, will be an essential direction for the next step of research. We are expected to build a more secure and stable large-scale network system through these efforts.

### Funding
This study was supported by the Huzhou Key Laboratory of IoT Intelligent System Integration Technology (No. 2022-21). The funders had no role in study design, data collection and analysis, decision to publish, or preparation of the manuscript.

### Grant Disclosures
The following grant information was disclosed by the authors:
Huzhou Key Laboratory of IoT Intelligent System Integration Technology: 2022-21.

### Competing Interests
The authors declare that they have no competing interests.

### Author Contributions
- Yanyan Jiang conceived and designed the experiments, performed the experiments, analyzed the data, performed the computation work, prepared figures and/or tables, authored or reviewed drafts of the article, and approved the final draft.

### Data Availability
The code and raw data are available in the Supplemental Files.

## Supplemental Information

Supplemental information for this article can be found online at http://dx.doi.org/10.7717/peerj-cs.2755#supplemental-information.

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
