# Peer review of "A study of trust mining algorithms for beacon nodes in large-scale network environments"

_PeerJ Computer Science, doi:10.7717/peerj-cs.2755_

## Round 0.1 · original submission · Major Revisions

Dear Authors,

Thank you for submitting your article on the trust degree mining algorithm for beacon nodes in large-scale network environments. Your work addresses an important challenge in node positioning and trust evaluation. While the research is promising, some areas could benefit from clarification and expansion. Specifically, we (academic editor and reviewers) recommend the following (in addition to the detailed comments from the reviewers)

define key terms like "trust degree," "influence," "seepage theory," and "security status."
Expand on the mathematical formulation of "distance error evaluation" and "probability function" used to calculate direct trust. Include equations, if applicable
Break long sentences into shorter ones for better readability and understanding
Suggest areas for future work,

·

Basic reporting

-There are instances of repetitive phrasing (e.g., "large-scale network environment" appears excessively).

- The results section is thorough and links back to hypotheses, but conclusions regarding scalability and dynamic network adaptability are speculative and not directly supported by the presented data.

Experimental design

- The experiments limit communication distance to 100 meters without discussing why this specific threshold was chosen.

- While Figures 5 and 6 present results, the underlying statistical analyses (e.g., mean, variance) are not disclosed.

- The python-publish-test.yml file focuses only on publishing workflows. There is no evidence of automated testing (unit, integration, or regression tests) integrated into the CI/CD pipeline to ensure the reliability of code changes.

Validity of the findings

-The paper claims that trust levels reflect network security, but no formal threat model is presented.

- The use of percolation theory assumes joint influence propagation is negligible, which may not hold in dense or highly interconnected networks.

Additional comments

This study develops a trust mining algorithm for beacon nodes in large-scale networks to improve node localization and security. By evaluating trust using percolation theory and integrating it with RSSI and DV-Hop optimization, the proposed method reduces positioning errors, enhances network reliability, and effectively identifies malicious node attacks. However, the paper suffers from the limitations listed below, which must be addressed before its reconsideration:

1- In Figure 5, the normalized positioning error improves with 50 beacon nodes. However, no performance evaluation is shown for scenarios with fewer than 10 beacon nodes!

2- While the algorithm outperforms RSSI and DV-Hop in Figures 5 and 6, no statistical significance is reported.

3- How would node mobility or environmental changes affect trust levels and positioning accuracy?

4- Figure 1 (Probability Allocation Function): The graph lacks labeled axes and a legend, which makes interpreting the categories of "normal," "uncertain," and "abnormal" probabilities unclear.

5- Equation (22) integrates node influence into the DV-Hop algorithm. Are there sensitivity analyses showing the impact of varying influence weights on the final positioning accuracy?

6- Figure 2 (DV-Hop Localization Process): The figure is overly simplified and does not clearly represent the transitions between phases.

7- Table 2 lists a 3-year node lifespan. What happens to the trust model when nodes are replaced?

8- While data availability is stated, no details about the raw data structure or preprocessing methods are given. Could these omissions affect the reproducibility of the trust mining and positioning results?

·

Basic reporting

The paper presents an innovative approach to improving the positioning accuracy and security of wireless sensor networks, particularly in large-scale environments such as industrial parks. The proposed method, based on trust mining of beacon nodes, offers significant improvements over existing algorithms like DV-Hop and RSSI, particularly in terms of accuracy and robustness against malicious attacks. This work contributes meaningfully to the field by addressing the challenges of node localization in complex networks. However, while the results are promising, a few areas could benefit from further clarification, refinement, or additional detail to strengthen the overall quality and impact of the paper. Below are some suggested revisions and comments that can help improve the paper.
1. The introduction should provide a clearer explanation of the problem being addressed and why current solutions (DV-Hop, RSSI) fall short. Adding a bit more context on the limitations of these algorithms will help the reader understand the need for your approach.
2. The problem definition is somewhat implicit. Consider restating it explicitly, focusing on both the positioning challenges and security issues with beacon nodes in wireless sensor networks.
3. Some references (e.g., [1], [3], [7]) could be more thoroughly discussed in the context of their limitations. Clarify how they relate to your work and differentiate your approach from previous ones more explicitly.

Experimental design

In Figure 5, the performance comparison with the DV-Hop and RSSI algorithms is useful, but it would benefit from a brief summary of why the new algorithm achieves better results. A table summarizing error rates across different methods could also help visualize the comparison.

The figure should include a brief caption that explains the key areas or sections of the park layout. How does this layout impact the choice of sensor nodes and the trust mining process?
It might be useful to explain why certain parameters were chosen (e.g., green percentage, size) and how they influence the performance of the irrigation system or sensor network.

Validity of the findings

It might be useful to explain why certain parameters were chosen (e.g., green percentage, size) and how they influence the performance of the irrigation system or sensor network.

In Section 3.2, while the concept of trust mining is introduced, more detail is needed on how the trust values are calculated and how this compares to other trust models. Clarify the trust degree calculation process.

The role of beacon nodes (A, B, C, D, E) in the network should be explained more clearly. Why are these nodes particularly important, and how are they selected? More context on their deployment would help.

The proposed algorithm is mentioned but not explained in enough detail. A step-by-step explanation of how the algorithm incorporates trust mining would make it easier for readers to understand the innovation behind it.

Additional comments

The paper contains quite a few grammatical errors that need attention. A careful review to correct these issues would improve the overall readability and professionalism of the work.

---

## Round 0.2 · accepted · Accept

Thank you for your re-submission, I am pleased to inform you that your manuscript is being recommended for publication based on input from experts and my opinion. Thank you.

·

Basic reporting

The authors addressed my comments.

Experimental design

The authors addressed my comments.

Validity of the findings

The authors addressed my comments.